# Enhanced Long LoRA Inspired Perceiver Architectures for Auto-Regressive Language Modeling

## Abstract

The Transformer architecture has revolutionized the Natural Language Processing field and is the backbone of Large Language Models (LLMs). The Transformer uses the attention mechanism that computes the pair-wise similarity between its input tokens to produce latent vectors that are able to understand the semantic meaning of the input text. One of the challenges in the Transformer architecture is the quadratic complexity of the attention mechanism that prohibits the efficient processing of long sequence lengths. While many recent research works have attempted to provide a reduction from $O(n^2)$ time complexity of attention to semi-linear complexity, it remains an unsolved problem in the sense of maintaining a high performance when such complexity is reduced. One of the important works in this respect is the Perceiver class of architectures that have demonstrated excellent performance while reducing the computation complexity. In this paper, we use the PerceiverAR that was proposed for Auto-Regressive modeling as a baseline, and provide three different architectural enhancements to it with varying computation overhead tradeoffs. Inspired by the recently proposed efficient attention computation approach of Long-LoRA, we then present an equally efficient Perceiver-based architecture (termed as Long LoRA Pereceiver - LLP) that can be used as the base architecture in LLMs instead of just a fine-tuning add-on. Our results on different benchmarks indicate impressive improvements compared to recent Transformer based models.

## 1 Introduction

The Transformer architecture has revolutionized the field of artificial intelligence, especially in Natural Language Processing (NLP) Vaswani (2017). The recent success of Large Language models such as ChatGPT Achiam et al. (2023), Gemini Team et al. (2023), Llama Touvron et al. (2023); Dubey et al. (2024), etc. with their comprehension and reasoning capabilities, is a testament to the effectiveness of the Transformer architecture. Prior to Transformers, deep Convolutional Neural Networks (CNNs) had demonstrated amazingly well results in computer vision applications, however, their performance does not show the same effectiveness when applied to NLP. One of the reasons CNNs are typically less effective in NLP is their limited receptive field, due in part to the convolution operation. The Transformer on the other hand, uses the attention mechanism. This operation measures the pairwise similarity between the words of the entire input sequence in order to comprehend it.

Consider the matrix $Q$, $K$ and $V$ containing rows representing the learnt position encoded (PE) embedding of each token in $d$ dimensions (i.e., $1 \times d$). Then the attention $A = \text{softmax}(QK^T)$ contains the dot product similarity of each input token with every other token in the input sequence. For an input sequence with $n$ tokens, $Q, K \in \mathbb{R}^{n \times d}$ and attention $A \in \mathbb{R}^{n \times n}$. The Transformer divides the attention calculation into parallel heads, to be able to refine the learning and comprehend the contextual meaning of the input sequence. Each head computes the attention on a portion of the embedding dimension. The output in each head is computed by further multiplying the attention $A$ with $V$. The canonical Transformer's operation can be summarized by the following equations Vaswani (2017),

first the output of the $i^{th}$ head is:

$$H_i = \text{softmax}(\frac{Q_i K_i^T}{\sqrt{d_k}})V_i = A_i V_i \tag{1}$$

where $d_k = \frac{d}{h}$ is the dimension of each head, $h$ is the number of heads in each layer and $H_i \in \mathbb{R}^{n \times d_k}$. The output in each Transformer layer $Z$, is obtained by catenating the output of all heads and transformed further by a projection matrix $W^o$:

$$Z_j = \text{catenate}(H_0, H_1, ..., H_{h-1})W^o \tag{2}$$

where $W^o \in \mathbb{R}^{d \times d}$ and $Zj \in \mathbb{R}^{n \times d}$ (the same dimensions as the input). A classification layer is added to the last layer which predicts the next token in autoregressive generation:

$$out = classification(Z_{p-1}(Z_{p-2}(...Z_0(embedding(x) + PE(x)))) \tag{3}$$

Skip connections and layer normalization are also used in each layer to stabilize the training of the Transformer.

For language models, if text generation is the goal, the model is trained in an autoregressive manner where it learns to predict the next token given an input sequence of tokens. In autoregressive generation, the previously predicted token becomes part of the next input sequence. For autoregressive NLP models, the training process can be made highly effective by masking the attention matrix in a triangular fashion so that future tokens are not visible. The triangular masking helps in creating more (input, output) training pairs, as from an input training sequence of size $n$, $n-1$ training pairs can be created by simply hiding the next token one at a time. For NLP classification, the masking of the attention is not needed, as the classification decision is made on the entire input sequence.

Since the attention computation in each head measures the pairwise similarity in the input sequence, its time complexity is $O(n^2)$ if the sequence length is $n$. With larger NLP models being created operating on longer sequence lengths, and with multiple heads in each layer and many layers, the computational costs of Transformer training is becoming an important issue. Considerable research effort is being put into making the attention mechanism more efficient since it is the dominant computation. Numerous research papers have proposed ideas to reduce the quadratic time complexity of attention to linear or sub quadratic complexity. Some of the important works in this respect include: TransformerXL Dai et al. (2019), Linformer Wang et al. (2020), Longformer Beltagy et al. (2020), Reformer Kitaev et al. (2020), Performer Choromanski et al. (2020), Long-Short Attention Zhu et al. (2021), Perceiver Hawthorne et al. (2022); Jaegle et al. (2021); Jaegle et al., among others. Recently State Space Models Gu et al. (2021); Fu et al.; Dao & Gu have drawn considerable research attention and provided impressive results in some domains. It is yet to be seen if they can be better alternatives for NLP generative models. Thus we limit our comparisons to transformer-based architectures in this work. We provide a brief background in the above related Transformer approaches towards achieving lower attention complexity. Then we elaborate on the PerceiverAR Hawthorne et al. (2022) architecture that we further enhance in this work.

## 2 RELATED WORK AND CONTRIBUTIONS

One of the important works that reduces the quadratic time complexity of attention to $O(n)$ is Linformer Wang et al. (2020). The authors of Linformer empirically observed that attention has low rank, and therefore can be approximated by a low rank matrix. To achieve this, the $Q$ and $V$ matrices $\in \mathbb{R}^{n \times d}$ are projected to lower dimensions by using a learnable projection along the sequence dimension. This results in $Q, V \in \mathbb{R}^{k \times d}$ where $k < n$. Consequently the attention is $A = QK^T \in \mathbb{R}^{n \times k}$. Since $k$ is fixed the attention complexity is $O(n)$. Note that the output $A \times V$ is still $\in \mathbb{R}^{n \times d}$, i.e., input and output dimensions match. Because Linformer compresses the information along the input sequence dimension during the projection process, it cannot be used in effective autoregressive training as the masking of attention for future tokens cannot be accomplished. However, for classification problems where masking of attention is not needed, the Linformer architecture is effective in reducing attention complexity. Another notable work in the area of reducing the attention complexity is Reformer Kitaev et al. (2020) which uses locality sensitive hashing. This reduces the attention complexity to $O(n\log(n))$.

A popular important architecture that allows autoregressive modeling and handles long input contexts efficiently is termed as TransformerXL Dai et al. (2019). It follows a different approach than trying to compress the input sequence. It divides the input sequence into segments and uses segment level recurrence. After processing one segment, the model reuses the hidden states from the last layer of that segment as the initial hidden state for the next segment. It also introduces relative positional encoding to allow it to handle longer contexts. TransformerXL presented excellent perplexity results on benchmarks. Its slight drawback is the recurrent mechanism that may lose information over long contexts. A mathematical approach to reducing the attention complexity is taken in Performer Choromanski et al. (2020) which uses random features and projections. The attention is decomposed into a product of non-linear functions of original query and key matrices referred to as random features. This allows the attention to be theoretically encoded more efficiently but the complexity of this encoding for long sequences may be higher.

In Longformer Beltagy et al. (2020) a different approach was employed which relies on sparse attention in order reduce its complexity. The Longformer authors proposed different sparse attention patterns such as sliding window, dilated sliding window attention and partially sparse global attention. In sliding window, only the nearby tokens are attended to, whereas in dilated sliding window, a diagonal partially sparse attention pattern is used. One architecture that partially relies on sliding window attention in handling long sequences is the long-short Transformer Zhu et al. (2021). Here the short attention refers to the sliding window while the long attention divides the entire context into small compressed segments. Both short and long attention are combined in the final attention. Since the long attention is based on compression, it may lose some important contextual information in an autoregressive generation.

Another approach to efficient attention was proposed in PerceiverAR Hawthorne et al. (2022). It accomplishes efficient handling of long contexts by diving the input sequence in two components of history and latent. The query matrix is computed on the latent part, whereas the key and value are computed on the entire context. This results in a cross attention in the first layer $\in \mathbb{R}^{l \times d}$ where $l$ is the latent size. The output from the first layer $\in \mathbb{R}^{l \times d}$. The remaining layers operate on the $l \times d$ which is smaller than the $n \times d$ size in a standard Transformer. For Transformers with many layers, the PerceiverAR approach is quite efficient because of the smaller size being operated upon after the first layer. Since the history component of the input is not used in autoregressive modeling, and since it also gets compressed into the latent part after the first layer, we attempt to improve upon these short coming via different architectural enhancements in this paper.

An important advancement in handling very long contexts has been recently proposed in LongLoRA Chen et al.. Even though it computes attention in a sliding window manner, it captures the entire context via a division of the attention heads into two groups. It performs a shift in the attention in the second group to propagate the attention information. We use this intriguing concept in a different way in enhancing the PerceiverAR architecture to accomplish same goals as LongLoRA in terms of attention efficiency.

1. We propose three enhancements to the PerceiverAR architecture to overcome the loss of information that gets compressed into the latent part after the first layer. Each enhancement has efficiency computation overhead tradeoffs.

2. Inspired by the ideas of LongLoRA, we present a simple overlapping PerceiverAR segmented architecture that achieves the computational efficiency of sliding window but with the entire context being available as the computation flows down the layers of the Transformer.

3. In addition to the efficient computation, the pairwise overlapping segment attention extracts more meaningful context resulting in a high performance architecture, which we empirically verify.

In the following section, we first elaborate on the PerceiverAR architecture before presenting enhancements to its design.

## 3 PERCEIVERAR ARCHITECTURE

The fundamental concept behind PerceiverAR is to split the input sequence into two components which we denote as the history and latent sequences. We denote the input as $x$ with corresponding sequence length $n$. After the tokenization and embedding is carried out, the input can be considered as composed of a history component and a latent component as: $x \in \mathbb{R}^{n \times d} = x_{history} || x_{latent}$,

where $||$ indicates the catenation of two components. We denote the history length as $h$ and the latent length as $l$. After the embedding operation, $history \in \mathbb{R}^{h \times d}$ and $latent \in \mathbb{R}^{l \times d}$. The PerceiverAR computes the query only on the latent part in the first layer of Transformer, while the key and values are computed on the entire sequence length of size $n$. Thus the attention computation in the first layer produces an output of dimension $\mathbb{R}^{l \times d}$:

$$Q_{latent} = W_q x_{latent} \in \mathbb{R}^{r \times d} \tag{4}$$

$$K = W_k x \in \mathbb{R}^{n \times d} \tag{5}$$

$$V = W_v x \in \mathbb{R}^{n \times d} \tag{6}$$

$$Output = [\text{softmax}(Q_{latent} K^T) V] = [AV] \in \mathbb{R}^{l \times d} \tag{7}$$

Since the output from first layer of Transformer $\in \mathbb{R}^{l \times d}$ the remaining layers do a normal attention on inputs of size $l$, without splitting the input into two parts as done in the first layer. For autoregressive training, an input of $n$ tokens is used to create $(n-1)$ training pairs, such that the expected output of first token is the second token, and the expected token of inputs up to $(m-1)$ tokens is the $m^{th}$ token. The PerceiverAR uses the history part as a fixed input, hence the autoregressive training can only be done on the latent part of input. Thus, to hide the future tokens in the training of the first layer, the upper triangular part of the attention matrix is set to $-\infty$. The remaining layers operate only on the input size of the latent length, so the triangularization of the attention matrix is done on the square matrix corresponding to the latent part.

The attention architecture of the PerceiverAR is shown in Figure 1a. The additional details of layer normalization, feed forward network and skip connections are omitted. The attention complexity of PerceiverAR in the first layer of Transformer is $O(l \times n)$ while the remaining layers have a complexity of $O(l^2)$. This provides a significant reduction in computation, especially when $l < n$ and many layers are used in the Transformer. While the PerceiverAR has been able to accomplish impressive results on NLP benchmarks Hawthorne et al. (2022). It has two main drawbacks which we address in this work:

1. **Latent Training Dependency** - The training for AutoRegressive generation can only use the latent part of the input. Therefore, more training is required to accomplish the same learning as a normal Transformer (provided the model does not overfit).

2. **Lossy History** - The history is implicitly compressed into the latent output of first layer and is not explicitly refined as in a normal Transformer via many attention layers.

We improve upon the above drawbacks and present three different enhancements for better utilization of the history component in the PerceiverAR.

## 4 PERCEIVERAR ENHANCEMENTS

The baseline PerceiverAR uses the history information explicitly only in the first layer by computing the key and values on the entire input sequence, while the query is computed only on the latent part of the input. To overcome the loss of history information in subsequent layers, we propose the first enhancement where each layer computes two attentions and correspondingly two outputs.

### 4.1 ENHANCED PERCEIVERAR ARCHITECTURE V1

In this enhancement, each layer performs two attentions. The first attention is the computation as the PerceiverAR baseline as given by Equations 4 through 7. The second attention is based on the history component of the input and also computes the attention in each layer. The second attention generates an additional output as follows:

$$Q_h = W_{qh} x_{history} \in \mathbb{R}^{h \times d} \tag{8}$$

$$K_h = W_{kh} x_{history} \in \mathbb{R}^{h \times d} \tag{9}$$

$$V_h = W_{kv} x_{history} \in \mathbb{R}^{h \times d} \tag{10}$$

where $x_{history}$ is the history component of the input.

$$Output_1 = [\text{softmax}(Q_h K_h^T) V_h] = [A_h V_h] \in \mathbb{R}^{h \times d} \tag{11}$$

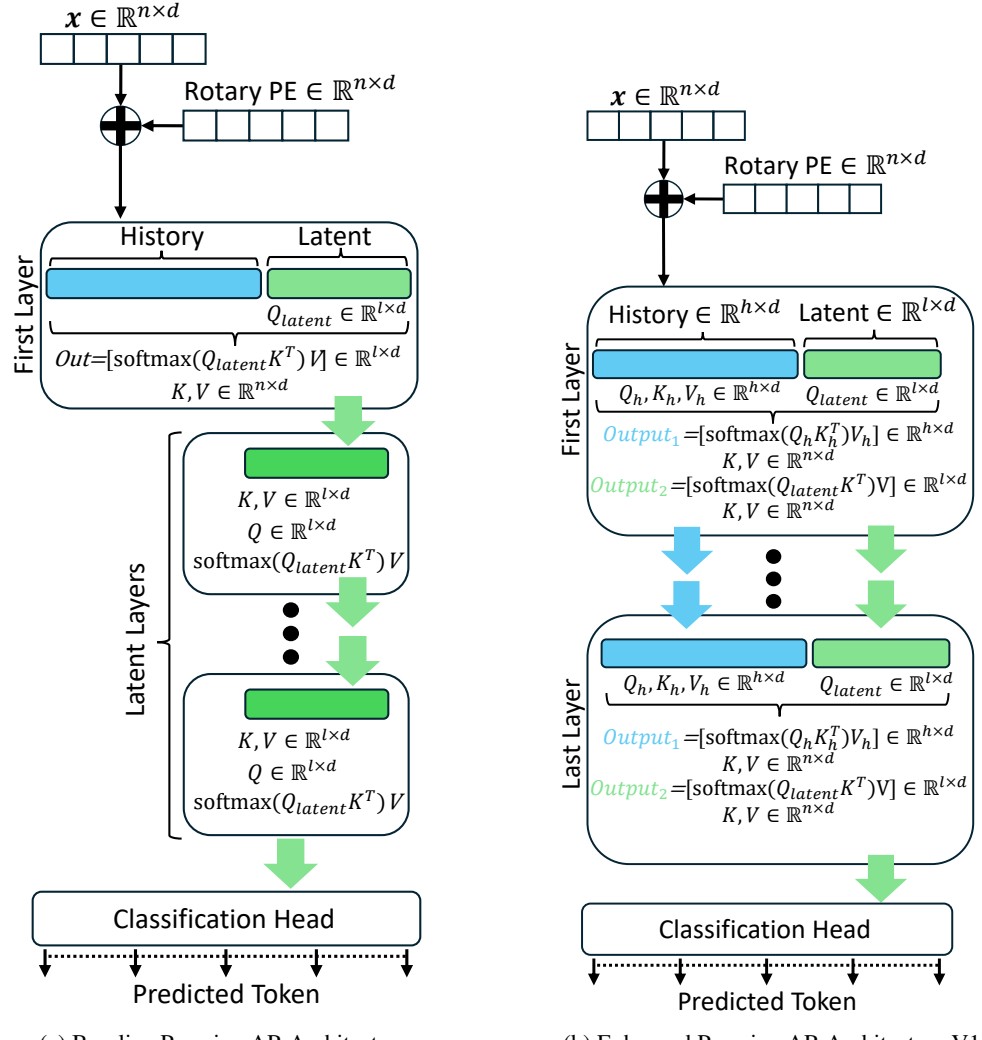

(a) Baseline PerceiverAR Architecture.  (b) Enhanced PerceiverAR Architecture V1.

Figure 1: Baseline Hawthorne et al. (2022) and Enhanced PerceiverAR Architecture V1.

$$Output_2 = [\text{softmax}(Q_{latent}K^T)V] = [A_{latent}V] \in \mathbb{R}^{l \times d} \tag{12}$$

Thus each layer in the enhanced V1 architecture is identical. The two outputs corresponding to the latent attention and the history attention are catenated to become the single output and the corresponding input for the subsequent layer. Note that no masking is used in the attention on the history part as this part is not used for autoregressive training. Only the latent attention uses masking. This enhanced architecture V1 is depicted in Figure 1b.

The overhead in this enhancement is the computation of the attention in the history component of the input. If the history length $h$ is larger than the latent length $l$, then this could be significantly more computations as compared to the baseline PerceiverAR where the subsequent layers after the first layer compute attention only on the latent part. To improve this drawback, we propose enhancement V2 as described in the next subsection.

## 4.2 ENHANCED PERCEIVERAR ARCHITECTURE V2

To make the history attention computation more efficient, we refine the V1 architecture by dividing the history component into smaller segments. Thus if the segment size $s$ in the history part is smaller than the latent length $l$, i.e., $s << l$, the overhead in the history computation in each layer is minimal. Note that the segment-wise attention is carried out within the same segment only, so if the segment size is $s$, the complexity of attention in each segment is $O(s^2)$. The output corresponding to each

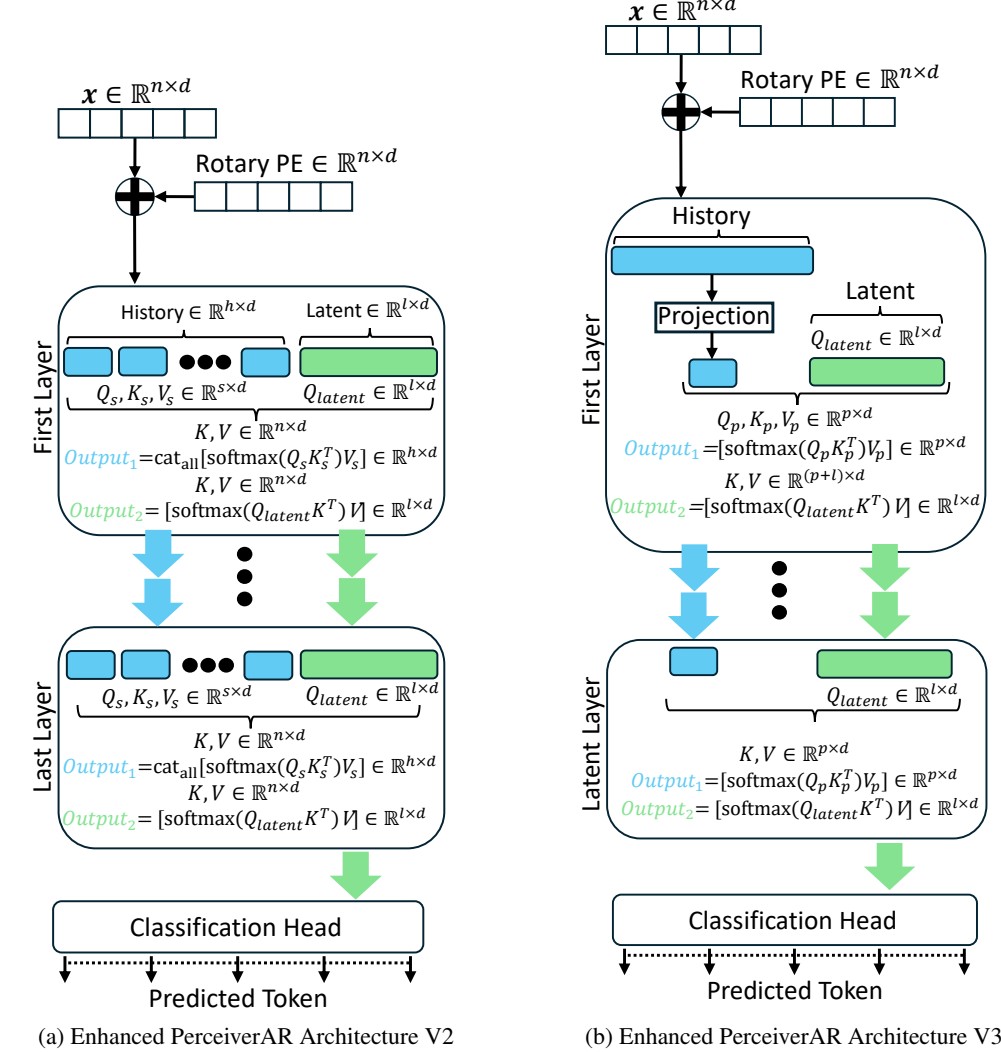

Figure 2: Other enhanced PerceiverAR Architectures (V2 and V3).

segment is catenated to become the history component of the output. The architecture of V2 version is shown in Figure 2a. We present one more enhancement for propagating the history part in a more efficient manner to the layers after the first layer.

### 4.3 ENHANCED PERCEIVERAR ARCHITECTURE V3

In this version, the first layer compresses the history part of the input by projecting it to a smaller length along the sequence dimension. This compresses the history information and this compressed history is then used and refined in all remaining layers. The attention and the computation of outputs in enhanced V3 architecture are shown visually in Figure 2b and given as follows. First, the the projection of the history to a compressed length $p$ is carried out only in the first layer:

$$x_{ph} = W_{ph}x_{history} \in \mathbb{R}^{p \times d} \tag{13}$$

All layers including the first layer implement the following:

$$Q_{ph} = W_{qh}x_{ph} \in \mathbb{R}^{p \times d} \tag{14}$$

$$K_{ph} = W_{kh}x_{ph} \in \mathbb{R}^{p \times d} \tag{15}$$

$$V_{ph} = W_{kv}x_{ph} \in \mathbb{R}^{p \times d} \tag{16}$$

$$Q_{latent} = W_{ql}x_{latent} \in \mathbb{R}^{l \times d} \tag{17}$$

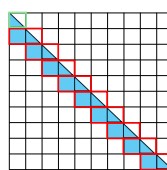

(a) PAR Block for a sequence of half segments in the first layer of the LLP architecture.

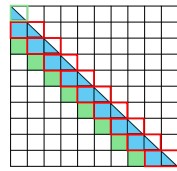

(b) Second layer of LLP architecture.

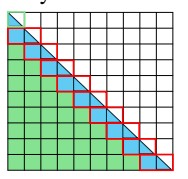

(c) After enough layers, previous segments information is implicitly available in PAR block's attention.

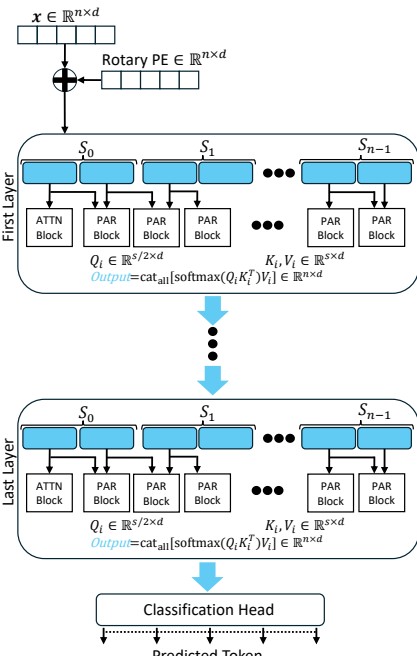

(d) LongLoRA Inspired PerceiverAR (LLP) Architecture.

Figure 3: In subfigures (a)-(c) the green rectangles indicates the ATTN block, the red rectangles indicates the PerceiverAR attention. The green blocks are not calculated but contain information because of the PAR block in the previous layer. The attention calculation is done only on the two blocks near the diagonal (indicated by red rectangles). Subfigure (d) shows the LLP architecture.

$$K = W_k(x_{ph}||x_{latent}) \in \mathbb{R}^{(p+l)\times d} \tag{18}$$

$$V = W_v(x_{ph}||x_{latent}) \in \mathbb{R}^{(p+l)\times d} \tag{19}$$

$$Output_1 = [\text{softmax}(Q_{ph}K_{ph}^T)V_{ph}] = [A_{ph}V_h] \in \mathbb{R}^{p\times d} \tag{20}$$

$$Output_2 = [\text{softmax}(Q_{latent}K^T)V] = [A_{latent}V] \in \mathbb{R}^{l\times d} \tag{21}$$

### 4.4 LONG LORA INSPIRED PERCEIVERAR (LLP) ARCHITECTURE

The complexity of computing attention is $O(n^2)$. One way to reduce this complexity is by dividing the input sequence into disjoint segments and only computing the attention in each segment itself. However, the disadvantage of such an approach is that there is information loss due to lack of information flow between segments. LongLoRA Chen et al. solves this problems via shifted sparse attention ($S^2$ Attn). In $S^2$ Attn, the sequence length is split into different groups and each group computes the attention individually. To support the information flow between different groups, the attention heads are divided in two halves. In the second half of the groups, the tokens are shifted by half the group size. This simple shift causes the information to be shared between neighboring groups. The primary application of LongLoRA was demonstrated in applying LoRA (Low Rank Adaptation) to the self attention layers in extending the sequence or the context length of existing LLMs. For example, LongLoRA can extend the context length of Llama-2 which is originally trained on a sequence length of 4K tokens by finetuning with data using 16K context length. This is accomplished by dividing the context length into 4K sequences. The $S^2$-Attn in LongLoRA will divide the attention heads in two groups, such that the first group conducts self-attention from 1st to 4096th tokens. For the second groups, the tokens are shifted by 2048, such that the attention in second group starts from 2049th token and ends at (4096+2028)=6144th token. This approach proposed by LongLoRA does not increase computation complexity, but allows information exchange between different groups. As the information flows through different layers of the Transformer, the information exchange expands to all the segments causing the net attention to become close to the standard full attention.

Inspired by the idea of LongLoRA's overlapping attention, we apply this concept to enhance the PerceiverAR design by dividing the input sequence into segments, such that the PerceiverAR allows communication of information from previous segment to the current segment. This is accomplished by first dividing the input sequence into disjoint segments. Next each segment is further divided into two halves, a history component and a latent component. The standard PerceiverAR computation is applied to consecutive pairs of half segments. Specifically, $Q$ is computed on the current half segment while computing $K$ and $V$ is done on the current and previous half segments. Thus the PerceiverAR operates upon overlapping segments, with the overlap amount being half the size of a segment. We term this approach as LongLoRA inspired Perceiver (LLP) architecture. It is depicted in Figure 3d. The ATTN block is a regular attention block which operates upon half of the first segment only. The attention equations governing the ATTN block with segment size $s$ are:

$$Q_{atn} = W_{qatn} x_{0:\frac{s}{2}} \in \mathbb{R}^{\frac{s}{2} \times d} \tag{22}$$

$$K_{atn} = W_{katn} x_{0:\frac{s}{2}} \in \mathbb{R}^{\frac{s}{2} \times d} \tag{23}$$

$$V_{atn} = W_{vatn} x_{0:\frac{s}{2}} \in \mathbb{R}^{\frac{s}{2} \times d} \tag{24}$$

$$Output_{atn} = [\text{softmax}(Q_{atn} K_{atn}^T) V_{atn}] = [A_{atn} V_{atn}] \in \mathbb{R}^{\frac{s}{2} \times d} \tag{25}$$

The PAR block performs the PerceiverAR operation on two consecutive half segments. The operations of the $i^{th}$ PAR block is given by the following set of equations:

$$Q_{par_i} = W_{qpart_i} x_{(\frac{s}{2})(i):(\frac{s}{2})(i+1)} \in \mathbb{R}^{\frac{s}{2} \times d} \tag{26}$$

$$K_{par_i} = W_{kpart_i} x_{(\frac{s}{2})(i-1):(\frac{s}{2})(i+1)} \in \mathbb{R}^{s \times d} \tag{27}$$

$$V_{par_i} = W_{vpart_i} x_{(\frac{s}{2})(i-1):(\frac{s}{2})(i+1)} \in \mathbb{R}^{s \times d} \tag{28}$$

$$Output_{par_i} = [\text{softmax}(Q_{par_i} K_{par_i}^T) V_{par_i}] = [A_{par_i} V_{par_i}] \in \mathbb{R}^{\frac{s}{2} \times d} \tag{29}$$

where in Equations 26 to 29, $i \in (1, 2 \cdot s_{total})$ and $s_{total}$ is the total number of segments. Each of the PAR blocks outputs data equal to half of the segment size i.e., $\mathbb{R}^{\frac{s}{2} \times d}$. All blocks use $Q \in \mathbb{R}^{\frac{s}{2} \times d}$ while $K, V \in \mathbb{R}^{s \times d}$ are computed on double the size i.e., on the current half segment and the previous half segment. Thus to implement autoregressive behavior masking is done on the $\frac{s}{2} \times \frac{s}{2}$ part of the attention in PAR block that is of size $\frac{s}{2} \times s$. The very first block in each layer is different and does a normal attention computation with triangular masking on the first half segment. This is done so that even the first half segment can be used in autoregressive implementation, unlike a normal PerceiverAR where the history part cannot be used in autoregressive training in terms of masking this part. All blocks in a layer output data of $\frac{s}{2} \times d$ size. These are then catenated to form an output of size $n \times d$. All layers in the LLP architecture are identical as shown in Figure 3d.

The main advantage of the proposed LLP architecture is that it provides identical efficiency in terms of computations as the LongLoRA design. Figure 3a 3b 3c shows the sparse attention pattern as it is computed in the first layer, and how each subsequent layer increases the attention receptive field (because of overlap of half segments in the PerceiverAR blocks). As shown in Figure 3c, after enough layers, the information from all previous segments is available to the PAR block as it calculates the attention on the two half segments. Thus we achieve a similar benefit in calculating the attention as Long LoRA. We also overcome the drawback of the PerceiverAR where the history component of the input sequence cannot be used in autoregressive training. In our LLP model, the entire input sequence can be used in autoregressive training, thus giving us the benefit of highly efficient attention computation, full sequence length auto regressive training, and availability of entire previous context as enough layers are used in the architecture.

## 5 EXPERIMENTAL RESULTS

We evaluate different enhanced PerceiverAR architectures on Wikitext-103 Merity et al. (2022) and PG-19 Rae et al. (2019) to test the perplexity of different models. In general, each model is trained under the same set of hyperparameters (batch size, total number of epochs and learning rate scheduler) to make the perplexity comparisons standardized.

**Model Configurations and Training** - In Table 1, the main experimental results are presented for three different model configurations (A, B and C). For all configurations, the history length in the Perceiver model is the sequence length subtracted from the latent length. In Configuration

| Configuration A | | | |
|---|---|---|---|
| | Baseline | Enhanced Arch V1 | Enhanced Arch V2 | Enhanced Arch V3 |
| Latent=256 | 63.8524 | 54.8534 | 55.4036 | 63.112 |
| Latent=512 | 43.2374 | 38.6215 | 38.7562 | 43.0051 |
| Latent=768 | 35.4916 | 32.9618 | 32.9865 | 33.4186 |
| Configuration B | | | |
| | Baseline | Enhanced Arch V1 | Enhanced Arch V2 | Enhanced Arch V3 |
| Latent=1024 | 31.902 | 30.3801 | 29.5097 | 29.9752 |
| Configuration C | | | |
| | Baseline | Enhanced Arch V1 | Enhanced Arch V2 | Enhanced Arch V3 |
| Latent=1024 | 28.2436 | 27.3821 | 27.1041 | 26.713 |

Table 1: Perplexity results for different architectures on the Wikitext-103 dataset. Configuration A, B and C represent different architecture variations and are fully detailed in Section 5.

| | LLP Model | | |
|---|---|---|---|
| | Segment Size=512 | Segment Size=256 | Segment Size=128 |
| Sequence Length=1024 | **25.3824** | 25.7923 | 25.8914 |
| Sequence Length=2048 | **20.0021** | 20.4536 | 20.7482 |

Table 2: Perplexity results for the LLP model with different segment sizes on Wikitext-103.

A, a sequence length of 1024 is used with an embedding size of 512, with 8 heads and 8 layers. Enhanced Arch V2 uses a segment size of 256 with latent sizes of 256 and 512. It uses a segment size of 128 when the latent length is 768. Enhanced Arch V3 compresses the history component to size 128. In Configuration B, the embedding size is 768, with 6 heads per layer and 6 layers. In this configuration, a sequence length of 2048 is used, with a latent length of 1024. Enhanced Arch V2 uses a segment size of 512. In this configuration, each model is trained for 200,000 iterations with a starting learning rate of $2^{-4}$. Lastly, in Configuration C an embedding size of 768 is used and there are 6 heads per layer with a total of 6 layers. A sequence length of 2048 and a latent length of 1024 is used. Enhanced Arch V2 uses a segment size of 512 and model V3 uses a compression size of 256. Models in this configuration are trained for 500,000 iterations.

**Discussion of Results:** From Table 1, it can be seen that Enhanced Arch V1 performs better than other enhanced models, and significantly better than the baseline PerceiverAR. This is due to the fact this enhancement carries the history component to all layers and keeps refining this information in each layer. Unlike the baseline model, where the history information is absorbed into the latent part after the first layer. Enhanced Arch V2 comes close to Enhanced Arch V1's performance, as it also carries the history information to all layers. However, in Enhanced Arch V2, the history is divided into segments, and the attention is only done within the segment. Computation efficiency wise, this is better than Enhanced Arch V1. Enhanced Arch V3 compresses the history information and uses it in each layer. It does not seem to improve the perplexity (lower is better) significantly as compared to the baseline. Because LLP model is different in its design, we present the results for it separately after comparing the baseline with other model enhancements.

In Tables 2 and 3, we present results on the Wikitext-103 and PG-19 datasets for the LLP models. We achieve better perplexity than the PercieverAR baseline with significantly smaller (half or less) model size. This can be attributed to the pairwise overlapping application of the PerceiverAR style computation. This results in appropriate information extraction, and propagation of information of the entire context, down the layers of the transformer. Tables 4 and 5 provide a comparison of our LLP model with other SOTA models, including the PercieverAR baseline model, on both the Wikitext-103 and PG-19 datasets. As can be seen from the results, our model achieves the lowest perplexity with the smallest model size.

# 6 CONCLUSION

Efficient computation of attention in transformer models is an important area of research with significant impact on the training budgets of large language models. In this work, we enhance the PerceiverAR architecture by examining its limitations in language modeling and providing different enhancements to improve its performance. One of the reasons, we focus on the PerceiverAR design is that it divides the input context into two components i.e., latent and history. This two level breakdown can be exploited further in a Long LoRA style efficient computation of attention without loss

| Model Architecture | LLP - 12 layers | LLP - 18 layers | LLP - 24 layers | PerceiverAR* 60 Layers |
|---|---|---|---|---|
| Model Size | 129.60 million | 172.12 million | 214.64 million | 974.6 million |
| Test Perplexity Wikitext-103 | 19.92 | 17.82 | **17.43** | 18.35 |
| Test Perplexity PG-19 | 21.89 | 20.42 | **18.83** | 28.9 |

Table 3: Perplexity results on Wikitext-103 and PG-19. All LLP models use 6 heads, an embedding dimension of 768, a sequence length of 2048 and a segment size of 256. PerceiverAR* indicates result cited from Hawthorne et al. (2022).

| Model Architecture | Model Size | Perplexity |
|---|---|---|
| LLP (ours) | 172.12 million | **17.82** |
| LLP (ours) | 87 million | **20.00** |
| xLSTM[7:1][‡] | 163.7 million | 21.47 |
| RWKV-4[‡] | 169.4 million | 22.33 |
| Mamba[‡] | 167.8 million | 22.49 |
| Llama[‡] | 162.2 million | 23.16 |
| H3 (Hungry Hungry Hippos)[Δ] | 125 million | 23.70 |
| Transformer-XL[†] | 151 million | 24.00 |
| ∞-Former** | 150 million | 24.22 |

Table 4: Comparison of perplexity results on Wikitext-103 with published architectures of similar model sizes. † is from Dai et al. (2019), ‡ is from Beck et al. (2024), ** is from Martins et al. (2021) and [Δ] is from Fu et al. (2022).

| Model Architecture | Model Size | Perplexity |
|---|---|---|
| LLP Transformer (ours) | 24 Layers – 214 million | **18.83** |
| Compressive Transformer* | 36 Layers – size unknown | 33.6 |
| Routing Transformer* | 22 Layers – 490 million | 33.2 |
| Transformer-XL[†] | 36 Layers – size unknown | 36.3 |
| Block Recurrent Transformer* | 24 Layers – 1.3 billion | 26.5 |

Table 5: Comparison of perplexity results for the LLP model and other models on PG-19 Dataset. * indicates results reported from Hutchins et al. (2022) and † indicates results reported from Dai et al. (2019).

of performance with respect to full attention. Similar to Long LoRA, where the different attention heads divide the input tokens into groups, and half the groups are shifted to propagate the attention, we equivalently divide the input into segments, and compute the PerceiverAR operation on a pair of consecutive overlapping segments. This overlap of half segments accomplishes the same goal as Long LoRA and causes the propagation of attention information from all previous contexts to the current segment. Our results indicate efficient computation with excellent perplexity. Finally, it should be noted that our Long LoRA inspired PerceiverAR based architecture can use the Long LoRA in the attention heads to further improve the language model performance.

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

# 7 APPENDIX

## 7.1 EFFICIENCY AND COMPLEXITY ANALYSES

In this subsection, we summarize the architectural attributes for different enhancements in this work. The number of calculation steps needed in the attention for different models is shown in Table 8. Each attention step indicates calculation of one attention entry. For example, in the canonical Transformer, the number of calculation steps in attention will be $n \times n$. As a comparison, the attention steps for different architectures are shown in Table 6 with sequence length $n = 4096$, layers $l = 48$, heads $h = 24$, segment size $s = 256$ and projection $p = 256$ .

As can be seen from Table 6, the LLP model is extremely efficient with only $6\%$ of the computation needed in attention with respect to full attention in a Transformer. It also performs the best due to its pairwise extraction of attention information and propagation of attention down the layers similar to Long LoRA.

| Model | Number of Calculation Steps in Attention | Percentage of Full Attention |
|---|---|---|
| PerceiverAR (baseline) | 4932 million | 25% |
| Enhanced Model V1 | 14495 million | 75% |
| Enhanced Model V2 | 9682 million | 50% |
| Enhanced Model V3 | 14858 million | 77% |
| Long LoRA Inspired Model (LLP) | 1226 million | 6% |
| Transformer (full attention) | 19327 million | 100% |

Table 6: Relative computation efficiency of attention for different models with 24 heads and 48 layers. Sequence Length = 4096.

| Model | Attention Design |
|---|---|
| PerceiverAR (baseline) | First Layer uses Q on latent length. K, V on entire sequence to compute attention. Output from first layer is latent size. Remaining layers use attention on latent size only. |
| Enhanced Model V1 | All Layers use Q on latent length. K, V on entire sequence to compute attention. All layers also use attention on the history part. The output from each layer is (history+latent) size. |
| Enhanced Model V2 | All Layers use Q on latent length. K, V on entire sequence to compute attention. History is divided into smaller segments. All layers use self attention on each individual segment in the history part. Output from each layer is (history+latent) size. |
| Enhanced Model V3 | First Layer uses Q on latent length. K, V on entire sequence to compute attention. First layer compresses the history to size p, then all remaining layers use attention on the compressed p size history, and compute Q on latent, and K,V on (p+latent) part to compute attention. Output in remaining layers is (p+latent) size. |
| Long LoRA Inspired Model (LLP) | Input is divided into segments. All layers perform PerceiverAR style attention on each segment with an overlap of s/2 for each attention computation. |

Table 7: Summary of the attributes of different enhanced architectures in this work.

## 7.2 ADDITIONAL EXPERIMENTAL RESULTS

We test our LLP architecture on the image part of the Long Range Arena (LRA) benchmark Tay et al.. In the image part of LRA, the CIFAR-10 dataset (referred to as sCIFAR-10) is treated as a sequence of grayscale pixels. Table 9 presents the results on image classification for different

| Model | Number of Calculation Steps in Attention |
|---|---|
| PerceiverAR (baseline) | $\left[\left[\frac{n}{2} \times n\right] + (l-1)\left[\frac{n}{2}\right]^2\right] \times h$ |
| Enhanced Model V1 | $\left[\left[\frac{n}{2} \times n\right] + \left[\frac{n}{2} \times \frac{n}{2}\right]\right] \times h \times l$ |
| Enhanced Model V2 | $\left[\left[\frac{\frac{n}{2}}{s} \times \frac{\frac{n}{2}}{s}\right] \times s + \left[\frac{n}{2} \times n\right]\right] \times h \times l$ |
| Enhanced Model V3 | $\left[\left[\frac{n}{2} \times n\right] + \left[\frac{n}{2} \times \left(p + \frac{n}{2}\right)\right] \times (l-1) + [p \times p] \times (l-1)\right] \times h$ |
| Long LoRA Inspired Model (LLP) | $\left[\left[\frac{s}{2} \times \frac{s}{2}\right] + \left[\frac{s}{2} \times s\right] \times \left[\left(\frac{n}{\frac{s}{2}}\right) - 1\right]\right] \times h \times l$ |
| Transformer – Full Attention | $(n \times n) \times h \times l$ |

Table 8: Number of Attention Calculation Steps in Different Models. n= sequence length, p = projection size in Model 3, s = segment size in Model V2 and LLP. l = number of layers, h = number of heads in each layer.

| Model Architecture | Test Accuracy |
|---|---|
| LLP Transformer(ours) – 12 layers | **64.42%** |
| LLP Transformer (ours) – 3 layers | **59.32%** |
| Transformer (with RoPE) | 51.32% |
| Transformer*** | 42.44% |
| Sparse Transformer*** | 44.24% |
| Performer*** | 42.77% |
| Longformer*** | 42.22% |
| Big Bird*** | 40.83% |

Table 9: Comparison of Transformer architectures on the sCIFAR-10 dataset. *** indicates results from Tay et al.

Transformer architectures. The Transformer architectures compared in Tay et al. used a smaller architecture with 3 layers and 4 heads in each layer. The embedding size used is 64, with a feed-forward network size of 128. For fair architecture comparison, we also used the same sizes for one of our LLP models. Our Transformer implementation uses Rotary Position Embedding (RoPE Su et al. (2024)) which yields better accuracy. Even though the state space models perform better than Transformer-based designs on sCIFAR-10, our LLP Transformer model produces the best known accuracy for Transformer-based architectures on the sCIFAR-10 dataset.

## 7.3 VISUAL EXPLANATION AND DISCUSSION OF THE LLP ALGORITHM

In the LLP algorithm, the PerceiverAR operation is performed on a pair of overlapping half segments $S_i^j$, where $i$ is the half segment number and $j$ is the layer number in each layer of Transformer. This is visually shown in Figure 4. The effective receptive attention field increases as computation progresses down the layers of Transformer. For example, the calculation of the output in segment $S4$ in layer 3 is denoted as $S_4^3$ and uses $S_4^2$ and $S_3^2$ from the previous layer (layer 2). $Q$ is computed on $S_4^2$ while $K$ and $V$ are computed on $S_3^2$ and $S_4^2$. Note that $S_3^2$ in turn uses $S_3^1$ and $S_3^1$ from the previous layer (layer 1), and $S_1^1$ further uses $S_0^0$ and $S_2^0$. The segment information accumulated by $S_4^3$ from previous segments is depicted with a yellow color in Figure 5. Thus, even though the attention computation is local with size $\frac{s}{2} \times s$ if the segment size is $s$, the result is that the propagation of information from all previous segments down the layers occurs. This is due to the fact that PerceiverAR style attention between pair of consecutive overlapping segments is performed in our LLP approach. The first half segment is treated as a special case (as it cannot form a pair with a previous segment), and normal full attention is carried out on it with upper triangular masking, to aid in autoregressive modeling.

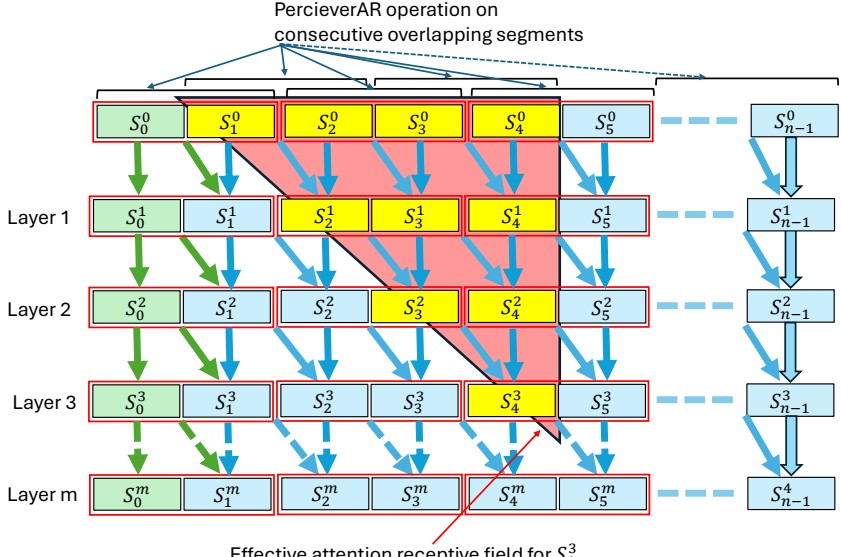

Figure 4: Visual depiction of the LLP Algorithm indicating increase in the attention receptive field as computation proceeds down the layers of Transformer.

**Advantages of the LLP Approach** - In the LLP approach, efficient attention computation without any loss of contextual information down the layers of the Transformer occurs. This is due to local pair wise segment attention, and overlapping segments in the LLP architecture. One reason, this approach performs better than full attention is that each layer in the early stages is learning to predict the next token with less information. For example, in Figure 4 the predicted tokens in $S_3^1$ (layer 1) only use the information from $S_2^0$ and $S_3^0$ and do not use context from $S_0^0$ and $S_1^0$. In layer 2, the computation of $S_3^2$ has information available from $S_0^1$, $S_0^2$ and $S_0^3$ (but not $S_0^0$). The following layer (layer 3) will have the information from all previous segments in computation of $S_3^3$. Thus, there is no loss of information in the next token prediction of any given segment in later layers, but initial layers are learning to predict the next tokens with less information. This is analogous to an implicit dropout in the earlier layers. This may be a contributing factor as to why our LLP model outperforms an equivalent full attention Transformer. In summary, the LLP approach combines efficiency and enhanced learning which results in a significantly improved Transformer architecture, as evidenced by the results on benchmarks.

```python
def LLPAttentionAlgorithm(x):
    # b=batch, n = sequence length, h = number of heads
    # compute qkv as in a regular transformer
    # split heads
    q, kv = map(lambda t: rearrange(t, 'b n (h d) -> (b h) n d', h = h), qkv)
    # split q, kv into segments
    qs = rearrange(q, 'c (s m) d ->c s m d', s = num_segments)
    kvs = rearrange(kv,'c (s m) d ->c s m d', s = num_segments)
    # catenate consecutive segments for kv
    fcat = lambda t : torch.cat([kvs[:,t,:,:],kvs[:,t+1,:,:]], dim=1).unsqueeze(dim=1)
    kvs_ccs = torch.cat([fcat(t) for t in range(0,self.num_segments-1)],dim=1)
    qs_ccs = qs[:,1:self.num_segments,:,:]
    lkv = self.norm(kvs_ccs)
    # compute attention in all segments
    attn = einsum('c s m d, c s n d -> c s m n', qs_ccs, kvs_ccs)

    # masking for autoregressive modeling
    m_size = attn.shape[-2]
    causal_mask = torch.ones(m_size, m_size, device = device).triu_(1).bool()
    attn[:,:,:,self.segment_len:].masked_fill_(causal_mask, mask_value)

    # attention in first segment - normal self attention
    attn0 = einsum('c s m d, c s n d->c s m n',qs[:,0:1,:,:],kvs[:,0:1,:,:])
    attn0.masked_fill_(causal_mask, mask_value)
    attn_0= attn0.softmax(dim=-1)
    attn_1 = attn.softmax(dim = -1)
    # apply dropout
    attnd_0 = self.attn_dropout(attn_0)
    attnd_1 = self.attn_dropout(attn_1)

    # generate outputs by multiplying attention with kvs
    out0 = einsum('c s i j, c s j d -> c s i d', attnd_0, kvs[:,0:1,:,:])
    out1 = einsum('c s i j, c s j d -> c s i d', attnd_1, kvs_ccs)

    out1 = rearrange(out1,'c s z d-> c (s z) d')
    out_1 = rearrange(out1, '(b h) n d -> b (n) (h d)', h = h) # combine heads
    out_1o = self.to_out(out_1) # project to compute output for the latent part
    out0 = rearrange(out0,'c s z d-> c (s z) d') # merge segments
    out_0 = rearrange(out0, '(b h) n d -> b (n) (h d)', h = h)
    out_0o = self.to_out_0(out_0) # history component
    out = torch.cat([out_0o, out_1o], dim = 1) # output both history and latent
    return out
```

Figure 5: Pseudocode for the LLP Attention in Pytorch Style

