# OpenReview forum: "Enhanced Long LoRA Inspired Perceiver Architectures for Auto-Regressive Language Modeling"
_ICLR.cc/2025/Conference — ICLR 2025 Conference Withdrawn Submission_

### Official Review · Reviewer_KRtd · 2024-11-02

**Soundness:** 1
**Presentation:** 1
**Contribution:** 1
**Rating:** 3
**Confidence:** 4

**Summary:**

This paper extends the PerceiverAR model by introducing three optimized versions aimed at improving the utilization of historical information. These enhancements address the issue of information loss that occurs due to compression into the latent space after the initial layer. Each enhancement involves trade-offs in computational efficiency. Experimental results on the Wikitext103 dataset demonstrate that the proposed model achieves lower perplexity compared to the PerceiverAR baseline.

**Strengths:**

This paper has implemented various optimization strategies based on PerceiverAR, resulting in lower perplexity on the Wikitext103 dataset compared to PerceiverAR.

**Weaknesses:**

1. **Limited experiments and baselines**. This paper only conducts experiments with PerceiverAR and the proposed model on the Wikitext-103 dataset. Furthermore, there is a lack of a substantial number of relevant baselines, such as Transformer[1], Transformer-XL[2], Compressive Transformer[3]. The paper only experiments with models of around 50M in size, which does not allow for validation of efficiency and effectiveness on larger models.
2. **Lack of key experiments**. The paper only presents theoretical computation times without offering practical runtime efficiency. It also lacks comparisons regarding model efficiency in real-world scenarios.
3. **Many grammar errors**. (1) line42- line44: "One of the reasons ~~for~~ (that) the CNNs are typically less effective in NLP is their limited receptive field because of the convolution operation."; (2) line68-line69: "Skip connections and layer normalization ~~is~~ (are) also used in each layer to stabilize the training of the Transformer."; (3) line137 - line138: "we attempt to improve upon these ~~short coming~~ (shortcomings) via different architectural enhancements in this paper"; and so on.


[1] Vaswani, A., Shazeer, N., Parmar, N., Uszkoreit, J., Jones, L., Gomez, A. N., Kaiser, L., and Polosukhin, I. Attention is all you need. In Proceedings of Neural Information Processing Systems (NeurIPS), 2017.

[2] Dai, Z., Yang, Z., Yang, Y., Carbonell, J., Le, Q. V., and Salakhutdinov, R. Transformer-XL: Attentive language models beyond a fixed-length context. In Proceedings of the Annual Meetings of the Association for Computational Linguistics (ACL), 2019.

[3] Rae, J. W., Potapenko, A., Jayakumar, S. M., Hillier, C., and Lillicrap, T. P. Compressive Transformers for long-range sequence modelling. In Proceedings of the International Conference on Learning Representations (ICLR), 2019.

**Questions:**

N/A

---

> ### Author Response · Authors · 2024-11-18
>
> In response to the reviewer's concerns:
>
> In response to comment “Limited experiments and baselines. This paper only conducts experiments with PerceiverAR and the proposed model on the Wikitext-103 dataset. Furthermore, there is a lack of a substantial number of relevant baselines, such as Transformer[1], Transformer-XL[2], Compressive Transformer[3]. The paper only experiments with models of around 50M in size, which does not allow for validation of efficiency and effectiveness on larger models.”
>
> We have added more results on different datasets as well as larger models. We have submitted a revised paper with greatly improved results section. We compare our LLP model on both Wikitext-103 and PG-19 datasets in the main body of the paper. We also evaluate our model on the image part of the Long Range Arena benchmark in the appendix of the revised submission. We urge the reviewer to please take a look at the revised submission and we believe that all of the above mentioned concerns have been taken care of in Tables 3, 4 and 5 in the revised submission.
>
> We compare our work with both Transformer-XL and Compressive transformer and achieve much better results.
>
> Table 3: Perplexity results on Wikitext-103 and PG-19. All LLP models use 6 heads, an embedding dimension of 768, a sequence length of 2048 and a segment size of 256. Full citations and experimental analyses are given in the update version of our paper:
>
> | Model Architecture           | LLP - 12 layers | LLP - 18 layers | LLP - 24 layers | PerceiverAR*  60 Layers |
> |------------------------------|-----------------|-----------------|-----------------|-------------------------|
> | Model size                   | 129.60 million  | 172.12 million  | 214.64 million  | 974.6 million           |
> | Test Perplexity Wikitext-103 | 19.92           | 17.82           | **17.43**           | 18.35                   |
> | Test Perplexity PG-19        | 21.89           | 20.42           | **18.83**           | 28.9                    |
>
> Table 4: Comparison of perplexity results on Wikitext-103 with published architectures of similar model sizes. Full citations and experimental analyses are given in the update version of our paper:
> | Model Architecture        | Model Size     | Perplexity |
> |:--------------------------|:---------------|:-----------|
> | LLP (ours)                | 172.12 million | **17.82**  |
> | LLP (ours)                | 87 million     | **20.00**  |
> | xLSTM\[7:1\]              | 163.7 million  | 21.47      |
> | RWKV-4                    | 169.4 million  | 22.33      |
> | Mamba                     | 167.8 million  | 22.49      |
> | Llama                     | 162.2 million  | 23.16      |
> | H3 (Hungry Hungry Hippos) | 125 million    | 23.70      |
> | Transformer-XL            | 151 million    | 24.00      |
> | Infinity-Former           | 150 million    | 24.22      |
>
> Table 5: Comparison of perplexity results for the LLP model and other models on PG-19 Dataset. Full citations and experimental analyses are given in the update version of our paper:
>
> | Model Architecture          | Model Size               | Perplexity |
> |-----------------------------|--------------------------|------------|
> | LLP (ours)                  | 24 Layers – 214 million  | **18.83**     |
> | Compressive Transformer     | 36 Layers – size unknown | 33.6       |
> | Routing Transformer         | 22 Layers – 490 million  | 33.2       |
> | Transformer-XL              | 36 Layers – size unknown | 36.3       |
> | Block Recurrent Transformer | 24 Layers – 1.3 billion  | 26.5       |
>
> Table 9: Comparison of Transformer architectures on the sCIFAR-10 dataset. Citation to relevant models and results are given in the Appendix.
> | Model Architecture                | Test Accuracy |
> |-----------------------------------|---------------|
> | LLP Transformer(ours) – 12 layers |**64.42%**        |
> | LLP Transformer (ours) – 3 layers | 59.32%        |
> | Transformer (with RoPE)           | 51.32%        |
> | Transformer                       | 42.44%        |
> | Sparse Transformer                | 44.24%        |
> | Performer                         | 42.77%        |
> | Longformer                        | 42.22%        |
> | Big Bird                          | 40.83%        |
>
>
> In response to the comment “Many grammatical errors”.
>
> We apologize for these errors due to submission deadline. Since then we have corrected and improved the write up in many places, including further clarifications and results in the appendix.
>
> We understand the reviewer’s feedback but believe that these concerns have been fully addressed in the revised submission.

---

> > ### Comment · Reviewer_KRtd · 2024-11-26
> >
> > Thank you for your response. Additional experiments and baselines help understand the effectiveness of the model. I have already changed the score. However, I believe a major revision of this paper is required to elaborate on the concerns raised. Thus, I do not recommend acceptance.

---

> > ### Comment · Reviewer_hbcj · 2024-11-26
> > **The score is maintained**
> >
> > Thanks for your response. The score is maintained

---

### Official Review · Reviewer_hS1B · 2024-11-03

**Soundness:** 3
**Presentation:** 2
**Contribution:** 3
**Rating:** 5
**Confidence:** 4

**Summary:**

The paper identifies two main drawbacks of the PerceiverAR architecture. The authors then propose three enhancements to the PerceiverAR architecture. The authors introduce the Long LoRA Inspired Perceiver (LLP) Architecture based on the concept of shifted sparse attention (S2 Attn) from LongLoRA. This approach splits the input sequence into overlapping segments and applies the PerceiverAR computation to consecutive pairs of half segments. This allows for efficient computation and information propagation throughout the entire context. The paper evaluates these enhanced architectures on the WikiText-103 dataset and compares their perplexity results. Model V1 performs the best among the enhancements, while Model V2 offers a more computationally efficient approach. The LLP architecture shows impressive perplexity results, indicating its effectiveness and computational efficiency. The authors conclude by highlighting the LLP architecture's benefits, achieving both efficient computation and impressive results due to its pairwise overlapping attention computation and information propagation. They suggest that the LLP architecture can be further enhanced by incorporating Long LoRA in the attention heads.

**Strengths:**

1. The paper provides a thorough overview of existing approaches for reducing attention complexity in Transformers.
2. The authors clearly identify the limitations of the PerceiverAR architecture, setting the stage for their proposed enhancements.
3. The authors propose three different enhancements, each with its own computational overhead trade-offs, providing flexibility for different use cases.
4. The LLP architecture offers an efficient and effective approach for handling long contexts by leveraging the concepts of LongLoRA and PerceiverAR.
5. The authors provide experimental results on a standard benchmark dataset, demonstrating the effectiveness of their proposed enhancements.
6. The paper discusses the performance of different architectures and analyzes the reasons behind the observed results.

**Weaknesses:**

1. The paper only evaluates the models on the WikiText-103 dataset. Evaluating the proposed architectures on other benchmarks would provide a more comprehensive understanding of their capabilities.
2. While the authors compare their enhancements with the baseline PerceiverAR, they don't compare them with other state-of-the-art models, which would strengthen the paper's claims.
3. The paper would benefit from more detailed ablation studies to understand the impact of different design choices in the proposed enhancements.

**Questions:**

1. How does the choice of segment size affect the performance and computational efficiency of the LLP architecture? Are there optimal segment sizes for different datasets and tasks?
2. How well do the proposed enhancements integrate with other techniques for reducing attention complexity, such as local attention or sparse attention patterns?
3. How well do the proposed architectures generalize to other NLP tasks beyond language modeling, such as machine translation or text summarization?

---

> ### Author Response · Authors · 2024-11-18
>
> We thank the reviewer for their insightful comments and offer the following response:
>
> “The paper only evaluates the models on the WikiText-103 dataset. Evaluating the proposed architectures on other benchmarks would provide a more comprehensive understanding of their capabilities”
> While the authors compare their enhancements with the baseline PerceiverAR, they don't compare them with other state-of-the-art models, which would strengthen the paper's claims.”
>
> We have added additional results in the revised submission, and provide results on the larger models on Wikitext-103 and PG-19 datasets in the main body of the paper. We provide additional results on the image part of Longe Range Arena benchmark in the Appendix. In all cases, our LLP model produces the best results when compared to SOTA models of similar sizes including Mamba, Llama and xLSTM among others.
>
> Table 3: Perplexity results on Wikitext-103 and PG-19. All LLP models use 6 heads, an embedding dimension of 768, a sequence length of 2048 and a segment size of 256. Full citations and experimental analyses are given in the update version of our paper:
>
> | Model Architecture           | LLP - 12 layers | LLP - 18 layers | LLP - 24 layers | PerceiverAR*  60 Layers |
> |------------------------------|-----------------|-----------------|-----------------|-------------------------|
> | Model size                   | 129.60 million  | 172.12 million  | 214.64 million  | 974.6 million           |
> | Test Perplexity Wikitext-103 | 19.92           | 17.82           | **17.43**           | 18.35                   |
> | Test Perplexity PG-19        | 21.89           | 20.42           | **18.83**           | 28.9                    |
>
> Table 4: Comparison of perplexity results on Wikitext-103 with published architectures of similar model sizes. Full citations and experimental analyses are given in the update version of our paper:
> | Model Architecture        | Model Size     | Perplexity |
> |:--------------------------|:---------------|:-----------|
> | LLP (ours)                | 172.12 million | **17.82**  |
> | LLP (ours)                | 87 million     | **20.00**  |
> | xLSTM\[7:1\]              | 163.7 million  | 21.47      |
> | RWKV-4                    | 169.4 million  | 22.33      |
> | Mamba                     | 167.8 million  | 22.49      |
> | Llama                     | 162.2 million  | 23.16      |
> | H3 (Hungry Hungry Hippos) | 125 million    | 23.70      |
> | Transformer-XL            | 151 million    | 24.00      |
> | Infinity-Former           | 150 million    | 24.22      |
>
> Table 5: Comparison of perplexity results for the LLP model and other models on PG-19 Dataset. Full citations and experimental analyses are given in the update version of our paper:
>
> | Model Architecture          | Model Size               | Perplexity |
> |-----------------------------|--------------------------|------------|
> | LLP (ours)                  | 24 Layers – 214 million  | **18.83**     |
> | Compressive Transformer     | 36 Layers – size unknown | 33.6       |
> | Routing Transformer         | 22 Layers – 490 million  | 33.2       |
> | Transformer-XL              | 36 Layers – size unknown | 36.3       |
> | Block Recurrent Transformer | 24 Layers – 1.3 billion  | 26.5       |
>
> Table 9: Comparison of Transformer architectures on the sCIFAR-10 dataset. Citation to relevant models and results are given in the Appendix.
> | Model Architecture                | Test Accuracy |
> |-----------------------------------|---------------|
> | LLP Transformer(ours) – 12 layers |**64.42%**        |
> | LLP Transformer (ours) – 3 layers | 59.32%        |
> | Transformer (with RoPE)           | 51.32%        |
> | Transformer                       | 42.44%        |
> | Sparse Transformer                | 44.24%        |
> | Performer                         | 42.77%        |
> | Longformer                        | 42.22%        |
> | Big Bird                          | 40.83%        |
>
>
> In response to the question “How does the choice of segment size affect the performance and computational efficiency of the LLP architecture? Are there optimal segment sizes for different datasets and tasks?”
>
> We experiment with the segment size on the wikitext-103 and sCIFAR-10 datasets. From performance point of view, we find that the  optimal segment size is dataset dependent. For example, on the sCIFAR-10 dataset where image pixels are treated as tokens to the transformer, a smaller segment size i.e., 64 produces better results (sequence length in this case is 1024). For the Wikitext-103 dataset, a segment size of 512 yields better perplexity results when the sequence length is 2048 or longer. We provide the exact computation efficiency equations in table 8 in the appendix. A smaller segment size is more memory and computation efficient, but of course may result in loss of performance due to more segment fragmentation.

---

> ### Author Response · Authors · 2024-11-18
> **Part two of the author's response**
>
> Furthermore, we would like to add:
>
>
> In response to the question “How well do the proposed enhancements integrate with other techniques for reducing attention complexity, such as local attention or sparse attention patterns?”
>
> Our LLP design is equivalent of the sliding window attention with very high sparsity as depicted in Figure 3 a). It can be further integrated with the LongLoRA style sparse attention inside each head as we comment on it in the conclusions. This is the focus of our follow up work.
>
> In response to the question “How well do the proposed architectures generalize to other NLP tasks beyond language modeling, such as machine translation or text summarization?”
>
> Since our architecture is ideal for autoregressive language modeling, it is well suited for machine translation as well as text translation as both tasks require generation of output based on a given context. For NLP classification, we may not have a performance advantage as contexts in such cases are typically small (e.g., sentiment analysis on tweets, or customer reviews).
>
> We thank the reviewer for taking the time for a detailed review and for insightful questions and comments.

---

> ### Comment · Reviewer_hS1B · 2024-11-26
>
> Thanks for the explanition and additional experiements. I would like to keep the score as it is relatively high.

---

### Official Review · Reviewer_hbcj · 2024-11-03

**Soundness:** 2
**Presentation:** 2
**Contribution:** 2
**Rating:** 3
**Confidence:** 5

**Summary:**

The paper describes 3 extensions to PerceiverAR architecture:
(1) adding history to latent state in each layer
(2) the same as (1) but the attention on history part is splitted into n small segments  of length s with segment-wise attention
(3) history projected in small length p
(4) is similar to (2) but segments are shifted in the half of heads like in longLora

The first part is very similar to encoder-decoder, where encoder has only 1 layer with extra cost ( h x h+ h x l), where h is history length and l is latent length. In the second case, the cost is reduced to n x s x s + h x l .
The 3rd option is equivalent to encoder with 1 layer and extra projection in the end.

**Strengths:**

The paper proposes 3 relatively minor variations of Perceiver AR + addition from longLora. The paper describes 3 extensions to PerceiverAR architecture:
(A) The first extension adds history to latent state in each layer.  This  is very similar to encoder-decoder, where encoder has only 1 layer with extra cost ( h x h+ h x l), where h is history length and l is latent length.
(B) The second extension is similar to (1) but the attention on history part is splitted into n small segments  of length s with segment-wise attention to reduce the cost  to n x s x s + h x l .
(C) 3rd extension is similar to (1) bur history is first to fixed  small segment p to reduce cost to pxp + p x l. The 3rd option is equivalent to encoder-decoder where encoder has 1 layer and extra projection in the end.
(LLP) Finally, authors add segments shifts to option B to  use overlapping segments . In this case whole sequence is used in auto-rergressive training.

In the experimental part of the paper, authors compare baseline PerceiverAR with  4 proposed options.  LLP gives the best result, but the comparison with original Perceiver-AR is missing.

**Weaknesses:**

The paper demonstrates that LLP is better than Perceiver based versions.
It is not clear
*  how different LLP is from the LongLora architecture or from LongFormer by Beltagy 2020 https://arxiv.org/pdf/2004.05150.
*  how LLP is related to Perceiver idea of compression long context into smaller  latent block:  I didn't find any latent block in the Fig. (3)

Experimental part:
the comparison of LLP with original Perceiver-AR is missing (see Table 10 in the original Perceiver AR paper: https://proceedings.mlr.press/v162/hawthorne22a/hawthorne22a.pdf )

Writing style.
I would suggest  to combine section 1 (Introduction  ) and section (2 Related work and Contribution and remove the explanation how regular Transformer attentionw works (lines 47-78) .

**Questions:**

Do you use latent block in LLP?
How LLP is different from the LongLora or from LongFormer by Beltagy 2020 https://arxiv.org/pdf/2004.05150?

---

> ### Author Response · Authors · 2024-11-18
>
> “how different LLP is from the LongLora architecture or from LongFormer by Beltagy 2020 https://arxiv.org/pdf/2004.05150”.
>
> Our LLP algorithm operates at the layer level where the input context is divided into segments, and the computation of output uses two consecutive overlapping segments (we visually clarify this further in the appendix in the revised paper in section 7.3). Q is computed on the current half segment , and Q,K are computed on the current and previous half segment.
>
> This PerceiverAR style attention computation accumulates the information from the previous half segment when the output is produced for the current half segment. As more layers are used, this information keeps expand such that the current segment has all the previous context (please see the explanation in section 7.3).
> LongLoRA on the other hand uses a sliding window attention in each head, and the heads are divided into two groups such that the second group shifts the input context. This also has the effect of accumulation of information as more layers are used. Our approach is inspired by the Long LoRA idea and that is why we term our model as Long LoRA inspired PerceiverAR i.e., LLP). To summarize, we have a different way of accumulating information in a layer. In our LLP model, all heads are identical and all layers do the same computations. In Long LoRA, heads in a layer operate in two different groups. As we mention in the conclusions, the two techniques (our LLP and Long LoRA)  used together have the potential to further combine the information in a more effective way. We plan to explore this in our follow up work.
>
>  “How different is LLP architecture from LongFormer?”
>
> Longformer uses local and global sliding window attention (or also dilated window attention). There is still sparsity in the approaches of the Longformer attention and if the next token prediction is dependent on this missing information, the performance is compromised. Our LLP model has the entire context available to it (down the layers) despite only doing equivalent of a sliding window attention in each layer. We do provide a comparison to the Longformer in Table 9 in our revised submission on the sCIFAR-10 dataset.
>
> “Experimental part: the comparison of LLP with original Perceiver-AR is missing (see Table 10 in the original Perceiver AR paper: https://proceedings.mlr.press/v162/hawthorne22a/hawthorne22a.pdf )”
>
> We have addressed this by adding additional results in revised paper (please see Table 3 in revised submission). As can be seen, our LLP model achieves better perplexity on both Wimitext-103 and PG-19 datasets with much smaller model size.
>
> Table 3: Perplexity results on Wikitext-103 and PG-19. All LLP models use 6 heads, an embedding dimension of 768, a sequence length of 2048 and a segment size of 256. Full citations and experimental analyses are given in the update version of our paper:
>
> | Model Architecture           | LLP - 12 layers | LLP - 18 layers | LLP - 24 layers | PerceiverAR*  60 Layers |
> |------------------------------|-----------------|-----------------|-----------------|-------------------------|
> | Model size                   | 129.60 million  | 172.12 million  | 214.64 million  | 974.6 million           |
> | Test Perplexity Wikitext-103 | 19.92           | 17.82           | **17.43**           | 18.35                   |
> | Test Perplexity PG-19        | 21.89           | 20.42           | **18.83**           | 28.9                    |
>
>
>
> In response to the question “Do you use latent block in LLP?”
> Yes, in each consecutive pair of half segments, the current half acts as the latent and the previous half is used as the history. For example, if the half segments are labeled as S0, S1, S2, S3, ….. In the [S0, S1] segment, S1 will used as the latent, and S0 as the history, then in the [S1,S2] segment, S2 will be used as the latent and S1 as the history, and so on. We also visually clarify this in the section 7.3 in the revised submission (Figure 4).
> We thank the reviewer for their time in reviewing the paper and asking for appropriate clarifications.

---

### Official Review · Reviewer_4nEs · 2024-11-04

**Soundness:** 1
**Presentation:** 1
**Contribution:** 2
**Rating:** 3
**Confidence:** 4

**Summary:**

This paper focuses on the quadratic complexity issue of attention by proposing three architectures that combine PerceiverAR with LongLoRA, providing a trade-off between latency and performance. Specifically, these methods enhance historical information through (1) additional history vector, (2) chunk-wise history vector, and (3) token pooling-wise history vector. The authors pretrain three enhanced PerceiverAR LM in GPT-2 small-size on WikiText-103, with a context window size of 1024. Experimental results indicate that, at the same latent size, the three enhanced PerceiverAR model achieve lower PPL than the original PerceiverAR.

**Strengths:**

- The paper thoroughly reviews a series of approaches aimed at optimizing the quadratic complexity of attention.

**Weaknesses:**

- The paper is somewhat disorganized, with an unclear main narrative and an excessive focus on related works. The three proposed methods appear somewhat trivial and lack core insights.
- The experimental section provides limited evidence of the effectiveness of the proposed methods, as it only compares against the original PerceiverAR model rather than full attention or other baselines discussed in the paper. Additionally, there is no comparison of performance on downstream tasks, nor sufficient ablation studies.
- The three PerceiverAR variants proposed involve trade-offs between performance and efficiency, but lack analysis and discussion on latency.
- The writing and experimental sections are logically unclear and lack critical information, requiring substantial revision, as detailed in Questions. The paper is not suitable for acceptance at ICLR in its current form, and I recommend resubmission after major revisions.

**Questions:**

- Format: The citation format is incorrect; `\cite{}` should be `\citep{}`.

---

> ### Author Response · Authors · 2024-11-18
>
> We appreciate the reviewer’s comments in helping us improve our paper.
>
> “The three proposed methods appear somewhat trivial and lack core insights.”
>
> We will like to clarify that the we provide three enhancements on the PerceiverAR baseline, as well as a significant improvement that is inspired by the Long LoRA design (we term this as the LLP model). Our LLP model produces excellent results as compared to current SOTA models (not just the PerceiverAR baseline).
>
> “The writing and experimental sections are logically unclear”
>
> We have improved the results section and added results on the larger models on Wikitext-103, as well as PG-19 datasets. We also have included results on the image part of the Long Range Arena (in the appendix). In all benchmarks, we achieve the best results of currently published results on similar sized Transformer architectures. We also have added tables in the results section to provide comparison to recently published results including Mamba, Llama, and xLSTM among other recent models.
>
> We do realize that there were typos and some of the sentences were perhaps not as clearly put together. We have gone through and made corrections. We have revised the writing in the experiment and results section.
>
> As for why our LLP model produces such good results: We have provided further explanations in the appendix (section 7.3). We urge the reviewer to please have a look and reconsider their score.
>
> Snapshot of additional results on other models and more datasets:
>
> Table 3: Perplexity results on Wikitext-103 and PG-19. All LLP models use 6 heads, an embedding dimension of 768, a sequence length of 2048 and a segment size of 256. Full citations and experimental analyses are given in the update version of our paper:
>
> | Model Architecture           | LLP - 12 layers | LLP - 18 layers | LLP - 24 layers | PerceiverAR*  60 Layers |
> |------------------------------|-----------------|-----------------|-----------------|-------------------------|
> | Model size                   | 129.60 million  | 172.12 million  | 214.64 million  | 974.6 million           |
> | Test Perplexity Wikitext-103 | 19.92           | 17.82           | 17.43           | 18.35                   |
> | Test Perplexity PG-19        | 21.89           | 20.42           | 18.83           | 28.9                    |
>
> Table 4: Comparison of perplexity results on Wikitext-103 with published architectures of similar model sizes. Full citations and experimental analyses are given in the update version of our paper:
> | Model Architecture        | Model Size     | Perplexity |
> |:--------------------------|:---------------|:-----------|
> | LLP (ours)                | 172.12 million | **17.82**  |
> | LLP (ours)                | 87 million     | **20.00**  |
> | xLSTM\[7:1\]              | 163.7 million  | 21.47      |
> | RWKV-4                    | 169.4 million  | 22.33      |
> | Mamba                     | 167.8 million  | 22.49      |
> | Llama                     | 162.2 million  | 23.16      |
> | H3 (Hungry Hungry Hippos) | 125 million    | 23.70      |
> | Transformer-XL            | 151 million    | 24.00      |
> | Infinity-Former           | 150 million    | 24.22      |
>
> Table 5: Comparison of perplexity results for the LLP model and other models on PG-19 Dataset. Full citations and experimental analyses are given in the update version of our paper:
>
> | Model Architecture          | Model Size               | Perplexity |
> |-----------------------------|--------------------------|------------|
> | LLP (ours)                  | 24 Layers – 214 million  | **18.83**     |
> | Compressive Transformer     | 36 Layers – size unknown | 33.6       |
> | Routing Transformer         | 22 Layers – 490 million  | 33.2       |
> | Transformer-XL              | 36 Layers – size unknown | 36.3       |
> | Block Recurrent Transformer | 24 Layers – 1.3 billion  | 26.5       |
>
> Table 9: Comparison of Transformer architectures on the sCIFAR-10 dataset. Citation to relevant models and results are given in the Appendix.
> | Model Architecture                | Test Accuracy |
> |-----------------------------------|---------------|
> | LLP Transformer(ours) – 12 layers |**64.42%**        |
> | LLP Transformer (ours) – 3 layers | 59.32%        |
> | Transformer (with RoPE)           | 51.32%        |
> | Transformer                       | 42.44%        |
> | Sparse Transformer                | 44.24%        |
> | Performer                         | 42.77%        |
> | Longformer                        | 42.22%        |
> | Big Bird                          | 40.83%        |

---

> ### Comment · Reviewer_4nEs · 2024-11-24
>
> Thank you for the authors' response. However, after carefully reviewing the additional experiments, the updated draft paper, and comments from other reviewers, I still believe that this paper requires significant revisions before being ready for acceptance at the conference. I'll keep my original score. Look forward to seeing your next version of the paper.

---

### Author Response · Authors · 2024-12-03
**General Comment to All Reviewers and AC**

We understand the reviewers concerns and feedback. We (the authors) maintain that we have addressed all major points brought up by the reviewers and would request reconsideration due to the following  specific points:

1. **Experimental Results** - Were requested by all reviewers. We have shown additional results on PG-19, Wikitext-103 and  sCIFAR-10. In ALL cases our architecture LLP outperforms existing SOTA models including xLSTM, Mamba, Llama, RWKV-4, Transformer-XL and H3, by significant margins.

2. **Writing Clarifications** - All reviewers requested writing clarifications. We have made significant changes to the paper and provided all requested edits (as can be seen in the updated pdf draft).

For the reasons above, we would greatly appreciate any additional consideration so that we can publish and share this novel work with the machine learning community.

---

### Note · Authors · 2025-01-29

I have read and agree with the venue's withdrawal policy on behalf of myself and my co-authors.